# A Long Journey of CICA-17 Quinoa Variety to Salinity Conditions in Egypt: Mineral Concentration in the Seeds

**DOI:** 10.3390/plants10020407

**Published:** 2021-02-22

**Authors:** Juan A. González, Leonardo Hinojosa, María I. Mercado, José-Luis Fernández-Turiel, Didier Bazile, Graciela I. Ponessa, Sayed Eisa, Daniela A. González, Marta Rejas, Sayed Hussin, Emad H. Abd El-Samad, Ahmed Abdel-Ati, Mohamed E. A. Ebrahim

**Affiliations:** 1Fundación Miguel Lillo, Instituto de Ecología, Comportamiento y Conservación, T4000 Tucumán, Argentina; 2Institute for Biodiversity and Ecosystem Dynamics (IBED), University of Amsterdam, 1012 WX Amsterdam, The Netherlands; l.a.hinojosasanchez2@uva.nl; 3Fundación Miguel Lillo, Instituto de Morfología Vegetal, T4000 Tucumán, Argentina; mimercado@lillo.org.ar (M.I.M.); giponessa@lillo.org.ar (G.I.P.); 4Geosciences Barcelona, CSIC, 08028 Barcelona, Spain; jlfernandez@geo3bcn.csic.es (J.-L.F.-T.); mrejas@geo3bcn.csic.es (M.R.); 5CIRAD, UMR SENS, 34398 Montpellier, France; didier.bazile@cirad.fr; 6SENS, CIRAD, IRD, University Paul Valery Montpellier 3, 34090 Montpellier, France; 7Faculty of Agriculture, Ain Shams University, Cairo 11672, Egypt; sayed_eisa@hotmail.com (S.E.); sayed_hussin@hotmail.com (S.H.); 8Instituto de Bioprospección y Fisiología Vegetal (INBIOFIV), Consejo Nacional de Investigaciones Científicas y Técnicas (CONICET), T4000 Tucumán, Argentina; danigonz37@gmail.com; 9Vegetable Research Department, Agricultural & Biological Research Division, National Research Centre, Giza 12611, Egypt; emadhassanein@hotmail.com; 10Plant Production Department, Ecology and Dry Land Agriculture Division, Desert Research Center, Cairo 11753, Egypt; dr.abdelati@gmail.com; 11General Organization for Agriculture Equalization Fund, Giza 12511, Egypt; mohamedelsawalhy@yahoo.com

**Keywords:** *Chenopodium quinoa* Wild., salinity, mineral concentration, food, extreme environment

## Abstract

Quinoa may be a promising alternative solution for arid regions, and it is necessary to test yield and mineral accumulation in grains under different soil types. Field experiments with *Chenopodium quinoa* (cv. CICA-17) were performed in Egypt in non-saline (electrical conductivity, 1.9 dS m^−1^) and saline (20 dS m^−1^) soils. Thirty-four chemical elements were studied in these crops. Results show different yields and mineral accumulations in the grains. Potassium (K), P, Mg, Ca, Na, Mn, and Fe are the main elements occurring in the quinoa grains, but their concentrations change between both soil types. Besides, soil salinity induced changes in the mineral pattern distribution among the different grain organs. Sodium was detected in the pericarp but not in other tissues. Pericarp structure may be a shield to prevent sodium entry to the underlying tissues but not for chloride, increasing its content in saline conditions. Under saline conditions, yield decreased to near 47%, and grain sizes greater than 1.68 mm were unfavored. Quinoa may serve as a complementary crop in the marginal lands of Egypt. It has an excellent nutrition perspective due to its mineral content and has a high potential to adapt to semi-arid and arid environments.

## 1. Introduction

Climate change is a reality, and we already see today its effects on the physiology, growth, and yield of field crops. For instance, the frequency of heatwaves has increased in large areas of the world, and precipitation changes have become more unpredictable [1]. Besides, the climate change effects and the bad agronomic practices have increased the saline soil areas. Salinity limits crop yields due to a reduction in photosynthesis, respiration, and protein synthesis. Around 7% of all land area in the world (1000 million ha) is affected by soil salinity, and more than 77 million ha from the arable area are affected by high salt contents [2,3].

Nevertheless, the main problem is that the principal crops are using plant species adapted to “old climatic conditions”. Hence, it is necessary to look for alternative crops or “new crops” to face the “new climatic conditions”. In this sense, it is crucial to consider some species that grew during millennia in mountain regions under extreme environmental conditions. Mountain plants, especially those adapted and cultivated in different altitudinal levels, can be crucial due to the gene pool that allowed these adaptations. In this scenario, quinoa (*Chenopodium quinoa* Willd.), which has grown throughout the Andes in South America for 5000 to 7000 years [4,5], can be considered a good option. During a long period of cultivation by the Aymaras and Inca populations, this crop was grown in different ecological zones, from sea level, in Chilean varieties [6], to 2000 to 4000 m above sea level (a.s.l.) along the Andes. Quinoa presents a C3 photosynthetic pathway according to anatomic and carbon isotope discrimination studies [6], with high photosynthetic assimilation and an intrinsic water use efficiency (iWUE) [7,8].

Several studies confirmed quinoa as an important source of nutritional components such as essential amino acids, fatty acids, minerals, soluble sugars, and bioactive components [9,10,11]. Furthermore, numerous reports in the field or lab conditions showed that quinoa is a species with high resilience to abiotic stress, including salinity, drought, high temperature, and ultraviolet B (UV-B) radiation [12,13,14]. Quinoa can tolerate very high salinity concentrations, producing a complete life cycle even at water salinities of 500–750 mM NaCl [15,16,17]. Thus, it can be grown in very marginal environments, for example, in North Africa, where soil salinization and drought are serious issues. The high nutritional value maintenance under different stresses makes quinoa an excellent crop to grow in the aforementioned marginal environments and face climate change. This quinoa tolerance to edaphic and harsh climatic conditions is related to this crop’s high diversity along the Andes. In effect, there are more than 16,000 quinoa accessions stored in different seed banks in 30 countries, most of which are concentrated in Bolivia and Peru [18,19,20]. These accessions include the five ecotypes classified by Tapia (2015) [21]: (i) Valley quinoa; (ii) Altiplano quinoa; (iii) Salar quinoa; (iv) Sea level quinoa; and (v) Subtropical quinoa. Quinoa accessions of the different ecotypes are considered multipurpose plants: the seeds and leaves can be used as food, the biomass can be used as animal feed or as a cover crop, the colorants and the saponin content can be used in pharmaceutical and agroindustry, and plantings can serve as a phytoremediation tool for environmental cleanup [22,23,24].

Tapia’s classification accepts an implicit fact: each ecotype can thrive in the environment in which it was adapted. However, quinoa has been introduced at higher latitudes as a complementary crop with good adaptation [25]. Currently, quinoa is cultivated and experimented on in almost 130 countries [19], including The United States [26,27], India [28], Italy [29], and Egypt [16], among others. Quinoa adaptation’s success is due to its high plasticity to reach places that differ from its original location managing the sowing dates, taking advantage of the environmental offer (basically temperature and light). One of the perfect examples of quinoa plasticity is the CICA-17 variety. It was selected at 3800 m a.s.l. in Cuzco-Peru from the local variety Amarilla de Maranganí at Centro de Investigaciones de Cultivos Andinos (CICA, Universidad Nacional de San Antonio Abad del Cusco, Peru). CICA-17 belongs to the Altiplano ecotype, and it is tolerant to cold temperatures, low precipitation, and high salinity conditions. CICA-17 quinoa was introduced in northwestern Argentina in 1996–1998 from the American and European Test of Quinoa conducted by FAO-CIP [30]. Nowadays, CICA-17 is the variety most used by small producers in Northwest Argentina and especially in arid high mountain valleys (above 2000 m a.s.l.) where the climate is desert type. This variety has been cultivated in Egypt for ten years because of its good adaptation to its marginal places. Egypt has a considerable extension in arid, semi-arid, and marginal lands that constrain classical crop productivity. In this scenario, quinoa is becoming a complementary crop of high nutritional value. CICA-17 has a notable yield (near 2000 kg ha^−1^) either in mountain valleys at 2000 m a.s.l. as in lowlands at 200 m a.s.l. in northwestern Argentina and Egypt [7,31,32]. Eisa et al. (2018) [33] and Ebrahim et al. [34] showed that the CICA-17 yield varied between 2000 and 3000 kg ha^−1^ in a marginal land at El-Fayoum oasis (Egypt).

CICA-17 quinoa can provide a new complementary crop for dry-saline lands. Still, it makes it necessary to study the mineral concentrations in different grain and seed organs, especially if different soil salinities influence these elements. Often, the mineral study in quinoa was focused on Na and K because of their relationship with the osmotic adjustment mechanisms that halophytes exhibit [35] or on the presence of a few minerals in different quinoa seed organs, and the abrasion effect on the Ca and K pericarp content [36]. All these approaches are essential to understand the physiological behavior of this promising species. It is also relevant to understand how these mineral contents can vary in different soils and climatic conditions in field conditions if quinoa is used as food in marginal lands. Nowadays, it is known that quinoa’s seeds and leaves are a significant source of major minerals (calcium, magnesium, potassium, phosphorus, sulfur, and sodium), trace elements (iron, cobalt, zinc, copper, and manganese), and ultratrace elements (chromium, lithium, arsenic, nickel, molybdenum, selenium, tin, and vanadium) and that their content varies according to the genotype and the place where the cultivation is carried out [33,36,37,38]. However, the detailed mineral composition in different quinoa ecotypes is still scarce and even more so is their spatial distribution. For example, Prado et al. [37] reported 18 minerals present in quinoa grains, while Konishi et al. [36] mentioned only six minerals. Hence, we investigated the grain yield and size, the occurrence and content of minerals and their spatial distribution in the different grain tissues, in the CICA-17 quinoa crop grown in field conditions on soils with different salinity in the marginal lands of the Egypt.

## 2. Results

### 2.1. Soil and Irrigation Water Analysis

Electrical conductivity (EC) and organic matter, Cl, Na, Mg, K, Ca, SO_4,_ and Fe contents are higher in saline soil (Table 1). These parameters increase 43, 24, 17, 9, 2.6, 4, and 1.3%, respectively, compared to the non-saline soil. Table 2 summarizes the water irrigation analysis for both cases.

### 2.2. Grain Yield

Soil salinity negatively affected grain yield. A reduction of close to 47% concerning the grain yield was obtained in saline conditions compared to the non-saline one (Figure 1).

### 2.3. Grain Weight and Sizes

The weight of 1000 seeds of CICA-17 quinoa decreased by 13% in saline conditions compared to non-saline conditions (Figure 2). On the other hand, the grain size distribution (A: <1.41 – >1.0 mm; B: <1.68 – >1.41 mm; C: <2.00 – >1.68 mm; D: ≥2 mm) showed that the two largest grain sizes were unfavored by salinity conditions (Figure 3).

### 2.4. Mineral Content in Quinoa Grains

The most important mineral elements with biological activities showed two trends. While phosphorus (P), magnesium (Mg), and sodium (Na) increased their content under saline conditions, silicon (Si), potassium (K), calcium (Ca), and iron (Fe) contents decreased (Table 3). The same increase or decrease were detected for the other elements except for Mn, Cu, and Co, which exhibited the same behavior under non-saline and saline conditions. 

### 2.5. Mineral Spatial Distribution on Quinoa Grains

Quinoa seed has different tissues (Figure 4). Using SEM-EDX analysis, we found that only the pericarp accumulated sodium in both non-saline and saline soils. Grains developed under saline conditions showed 6.3 times greater sodium content than those detected under non-saline conditions (Table 4, Figure 5). Chlorine, P, and Br also increased significantly in pericarp in saline conditions. By contrast, soil salinity led to a decrease in Ca, Al, Fe, Cu, and Si. Meanwhile, potassium was located mainly in the pericarp without significant differences due to the soil salinity condition. In the endosperm and the perisperm, the K content increased in saline soil (2.5 and 2.2 times, respectively) while its concentration was more significant (2.7 times) in the embryo in non-saline soil. The other elements studied did not change significantly.

The Mg content increased significantly in the endosperm under saline soil, while Cl content increased in embryo, and perisperm and decreased in the endosperm. Sulfur content differed between soil salinities. It was stored mainly in the embryo in non-saline conditions, meanwhile, it accumulated more and perisperm in saline soil (6.3 and 3 times, respectively). Nitrogen was detected only in the cotyledons, endosperm, and perisperm, reducing their content in the last two tissues under salinity conditions. Silicon, Fe, Br, Al, and Cu were below the detection limit for embryo, endosperm, and perisperm as for Na, they are detected only in pericarp.

## 3. Discussion

The high salinity soils of Sah El Tina have not produced harmful effects on the quinoa crop cultivation (cv. CICA-17). As a halophyte, the plant displayed a series of physiological and morphological adaptations that allowed it to complete its cycle [35,36,37,38,39]. Saline soils affect grain yield and seed size, according to previous research [40,41]. In our case, we detected a decrease near 47% compared to the yield get in non-saline conditions. The weight of 1000 seeds decreased by 13% in saline soil, and only smaller grains (< 1.68 mm) were favored. From a commercial point of view, this finding is also essential because markets prefer large grains instead of small ones.

Regardless of the soil’s saline conditions where quinoa was grown, the most abundant minerals in the quinoa grains were Si, K, P, Mg, Ca, Na, Mn, Fe, Cu, Al, and Zn. However, an increase in P, Na, and Mg contents was observed due to soil salinity. Sodium, an essential mineral in the cell ionic balance, was detected only in the pericarp and not in other tissues (embryo, endosperm, and perisperm). It is evident that the pericarp structure (with different cell layers) is a shield to prevent the entry of sodium to the underlying tissues, but not for the chlorides that increased in saline conditions. Contradictorily, the mineral content of Mg and Mn was reduced by salinity in saline-sodic soils in Greece [42]. However, the salinity conditions of that field experiment are low (6.5 dS m^−1^) in comparison to our experiment (26 dS m^−1^) (Table 1). It is necessary to consider that 26 dS/m is the starting value of the saline soil’s electrical conductivity (EC). It probably increases during the life cycle because of the soil and water quality used (with high EC). So we can assume that EC is further increased during cultivation, and the seeds were produced under more significant saline stress conditions than the starting one. This hypothesis must be verified in future studies in the field.

Regarding the occurrence of certain minerals in quinoa, it is necessary to consider whether the analyses are performed on grains (pericarp + embryo) or only in seeds (without pericarp). The desaponification process removes the pericarp and probably all the elements specifically present in this tissue (Table 4) This feature is mentioned in many cases where the analysis was performed on flour, but it is not clear whether the grains used contained the pericarp or not. Our results showed that Na was only in the pericarp, Mg was present in the pericarp, embryo, endosperm, and perisperm. While S was found in all the tissues, P was only absent in the perisperm. All these features should be considered to prepare quinoa-based foods since the product will not have the same mineral composition based on whether or not it is desaponified. It is important to point out that saponins are present in different quinoa organs (leaves, flowers, fruits, and seeds), especially in seed coats (pericarp). Saponins must be removed by different methods (physical or chemical) to avoid conferring a bitter flavor to the quinoa products. In general, saponin concentration ranges from 0.01 to 5% on a dry weight basis [43,44]. There are no data concerning saponin concentration in the quinoa crop of Egypt. However, González et al. [45] observed that saponin content in CICA -17, grown in desert climatic conditions in a high valley in Northwest Argentina, varied from 2.3 to 6.9% according to different nitrogen treatments. Considering that saponins can increase their concentration under saline condition [29], we can conclude that desaponification must be a necessary process before consuming quinoa either as grain or as flour because the maximum acceptable level of saponin in quinoa for human consumption varies from 0.06 to 0.12% [46,47]. Besides, desaponification removes sodium from the grain, avoiding its potential negative effects when consumed.

Except for Sn, Ni, and As, already found by Prado et al. (2014), the other ultratrace mineral elements (Rb, Sn, Th, Nd, Pr, Nb, Sm, Ni, Y, La, Ce, As, Ti, Ge, V, Zr, Ga, Zn, Ba, and Pb) were first detected in quinoa in both non-saline and saline conditions. The presence of Cs was reported [48], but it was observed in aboveground parts of plants (stem and leaves). For many trace and ultratrace minerals detected in quinoa, the cellular level function is unknown, and their presence is only a passive accumulation from the soil and irrigation water. The role of Cr, Li, Si, Ni in human metabolism was already demonstrated [49,50]. Chromium participates in protein transport and improves diabetes [51], while Li is an essential element for regulating the central nervous system [52]. Following our results, Li has also been found in quinoa and amaranth (*Amaranthus caudatus*) consumed as food in the Northwest of Argentina [53]. Silicon is essential for Ca assimilation, the formation of new cells, and tissue nutrition [54], and Ni is necessary for the proper functioning of the pancreas [55]. Regarding arsenic, which is dangerous in high concentrations, several studies suggest too that it probably plays a physiological role in the metabolism of methionine, acting as an effector of acid metabolism amino sulfur [56]. Aluminum is typically considered a toxic element, but some studies in vitro suggest that this element plays an essential role in different biological systems (e.g., DNA synthesis stimulation or bone formation) [57].

In summary, quinoa can be considered a source of minerals as Cr, Li, Si, Ni, As, and Al concerning nutritious food and health. Besides, considering that quinoa foliage can accumulate some minerals such Ni, Cr, Cu, and Cd [24] and the hyperaccumulation of heavy metals in roots [58], we hypothesize that this species may be a good alternative for the remediation of contaminated soils.

## 4. Materials and Methods

### 4.1. Plant Material, Site Description, and Experimental Design

Grains of CICA-17 quinoa cultivar were selected for this study. Field experiments were conducted in 2015/2016 (mid-November to end-March) in two places in Egypt. One was in the Sahl El-Tina plain (named saline in this work), located in the northwestern coast of Sinai Peninsula (31° 02′ N, 32° 35′ E), and other in the Experimental Station (named non-saline) of Ain Shams University, Cairo (30° 03′ N, 31° 14′ E). Both places are arid, with an average annual precipitation of 60 mm yr^−1^ (Sahl El-Tina) and 20 mm yr^−1^ (Cairo), with rainfall concentrated between October and April. The average monthly temperature was 20.5, 12.8, 16.7 °C and 21.7, 12.1, 16.9 °C for maximum, minimum, and mean temperature in saline and non-saline places during the crop cycle, respectively. Representative soil samples in both locations were collected in the center of each plot at 0.60 m depth. Soil samples (6 in total) were obtained with a soil borer. Samples were mixed in the lab, and physical and chemical analyses were performed according to the standard methods published by Page et al. (1982) [59]. Results are listed in Table 1.

Experimental soils were prepared, including the construction of ridges. Compost with a rate of 8 t ha^−1^ and phosphorus at a 120 kg P_2_O_5_ ha^−1^ was added during the land preparation. Nitrogen was added as side-dressing at 160 kg N ha^−1^ in two equal rates after 30 and 51 days from the sowing date. Potassium was added at a rate of 140 kg K_2_O ha^−1^ at the flowering stage. Seeds of quinoa were sterilized with sodium hypochlorite solution (5% active chloride) for 10 minutes and then washed with distilled water several times and dried with tissue paper before planting. We sowed about ten seeds per hill to ensure germination. A complete randomized block design with six replicates (experimental plots) was used, with an average of 18 m^2^ for each plot (6 ridges with 5 m length and 0.6 m width). After four weeks from the sowing date, the seedlings were thinned to two or three seedlings per hill. Local sources provided irrigation water in both places, and Table 2 lists their chemical properties. A detailed description of the procedure is in Eisa et al. (2017) [16].

### 4.2. Yield Components 

Ten quinoa plants in each experimental plot were cut and air-dried for 7–10 days at the harvesting stage. The dried panicles were threshed by hand. Subsequently, we determined the grain yield (kg ha^−1^), the seed sizes, and the weight of 1000 seeds (g). The percentage of grain sizes was determined with sieves of different meshes.

### 4.3. Mineral Content

The grain samples were dried in an oven at 65 °C to constant weight. The dried samples were ground into a fine powder, passed through a 60-mesh sieve, and ashed (electric oven at 575 °C for 16 h). After cooling, ash subsamples (0.1 g) were digested with 10 mL of HF/HClO_4_/HNO_3_ (5.0/2.5/2.5, *v*/*v*) mixture in a closed teflon vessel (Savillex, Canada) at 90 °C for at least 12 h. Once digestion finished, the remaining acid was evaporated on a hotplate to incipient dryness. Next, 1 mL of HNO_3_ was added twice and was evaporated again to incipient dryness. Residual sediment was dissolved in HNO_3_ (1 mL) and transferred into a 100 mL volumetric flask. The digestion vessel was rinsed with deionized water several times, and washing water was also transferred to the volumetric flask. Flask volume was made up with deionized water and then used for chemical element analysis. A total of 49 elements, including major (e.g., Ca, K, Mg, Na, and P), minor (e.g., Zn, Fe, Cu. Mn, Co, and Na), and ultratrace mineral elements (e.g., Cr, Li, As, Ni. Mo, Se, Sn, and V) were quantitatively determined by high resolution inductively coupled plasma mass spectrometry (Element XR HR-ICP-MS, Thermo Scientific, Germany) at the labGEOTOP of institute Geosciences Barcelona (Spanish Research Council, CSIC, Barcelona, Spain) [37]. Quality control of element determinations was carried out using internal standards and control samples of known composition for each analyzed element (BDH Chemical, England). Of the 49 elements analyzed, only 34 were listed because the rest (15 elements) were lower than the equipment’s detection limit. Element concentration (mean of 3 repetitions) was expressed on a dry weight (dw) basis (Table 1). The accuracy and precision of analytical determinations by HR-ICP-MS were lower than 10%.

### 4.4. Mineral Spatial Distribution

To determine the spatial distributions and relative abundances of C, O, N, Na, K, Ca, Mg, P, Cl, Cu, Br, Al, Fe, Si, and S, three samples of achenes per saline and non-saline locations were analyzed by scanning electron microscopy (SEM, Supra55VP) coupled to an energy dispersive X-ray analyzer (EDX). SEM-EDX conditions were −10 °C, sample chamber pressure of −50 Pa, accelerating voltage of 20 kV). These analyses were conducted at CIME facilities (Integral Center of Electronic Microscopy of CONICET-UNT, Tucumán, Argentina). 

Complete grains (to study pericarps), seeds with manually removed pericarp (to expose the seed coat), and median longitudinal sections of whole grains (cut with a blade) were mounted and fixed with a commercial adhesive on sample plates. Maps of mineral distribution were made in the median longitudinal sections. Semi-quantitative analyses of mineral abundance were obtained from equivalent areas of different tissues or organs (pericarp, episperm, endosperm, perisperm, embryo, and cotyledon mesophyll).

### 4.5. Statistical Analysis

The mean values between treatments per tissue/organs were compared using the *t*-test at *p* ≤ 0.05 level of probability (statistical package SPSS Inc., version 11.0, Chicago, IL, USA).

## 5. Conclusions

This study demonstrated the high resilience of the CICA-17 variety of quinoa to two contrasting edaphic situations in Egypt. It was clear that mineral concentration can change when interacting with different soil types, irrigation waters, and other environmental conditions. Soil salinity induces substantial changes in the distribution of minerals in the different grain tissues. Although quinoa is a significant source of major minerals such as K, P, Mg, Ca, and Na, the contents of Si, Ni, Cr, and Li may be important for their healthy behavior in human metabolism. The detection for the first time of 17 elements (Rb, Th, Nd, Pr, Nb, Sm, Y, La, Ce, Ti, Ge, V, Zr, Ga, Zn, Ba, and Pb) in quinoa seed add a new value to quinoa crops as a potential application for phytoremediation processes. From our results, we can conclude that cv. CICA-17 may be a complementary crop in the marginal lands of high salinity in Egypt and the Mediterranean region, showing high potential concerning healthy food and environmental issues as phytoremediation.

## Figures and Tables

**Figure 1 plants-10-00407-f001:**
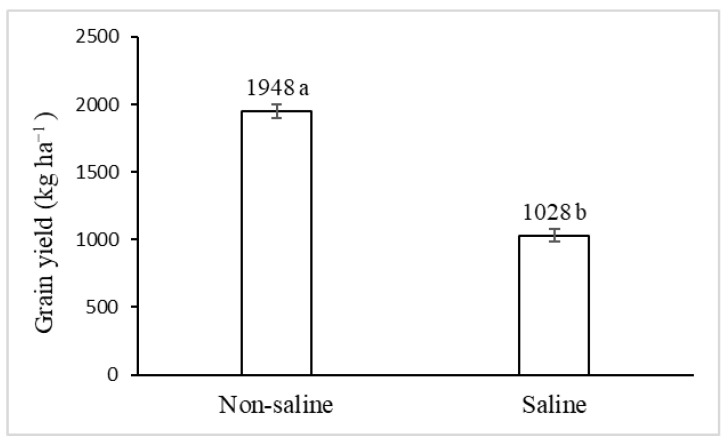
Effect of non-saline and saline soils on grain yield (mean ± SE) on *C. quinoa* cv. CICA-17. Different letters above columns indicate significant differences between means at *p* < 0.05, according to the *t*-test.

**Figure 2 plants-10-00407-f002:**
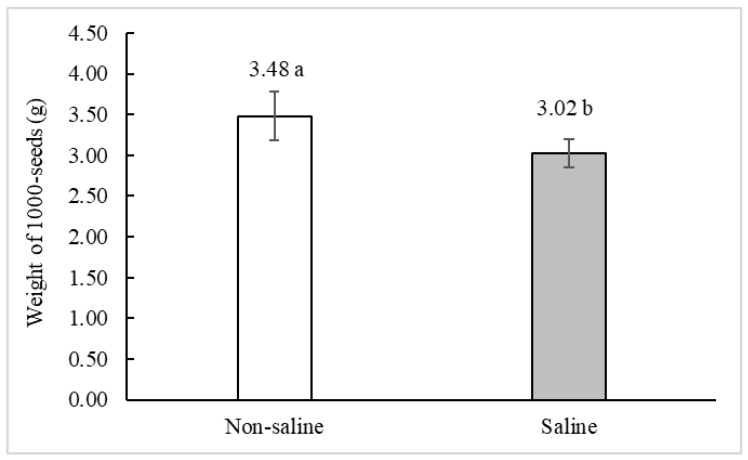
Effects of non-saline and saline soils on the weight (mean ± SE) of 1000-seeds of *C. quinoa* cv. CICA-17 plants. Significant differences between means (*p* ≤ 0.05) are indicated by different letters above columns according to the *t*-test.

**Figure 3 plants-10-00407-f003:**
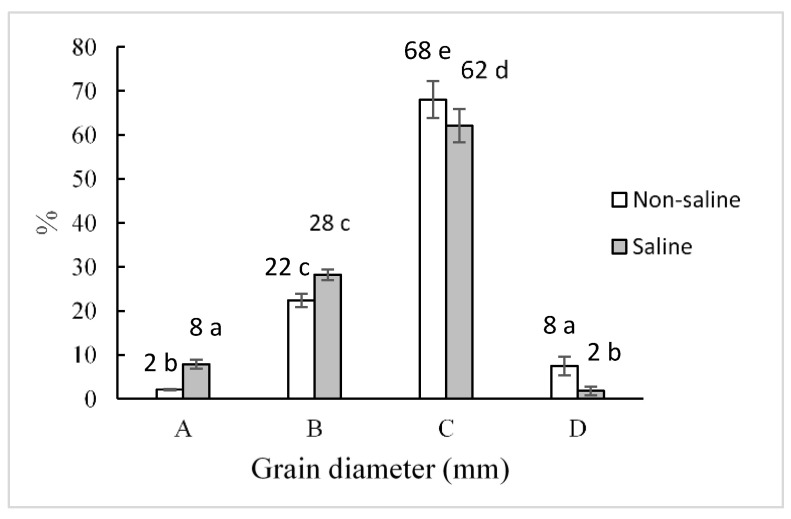
Seeds diameter distribution of *C. quinoa* cv. CICA-17 obtained in non-saline and saline soils. All the differences were significant (*p* ≤ 0.05) according to the *t*-test. A: <1.41 – >1.0 mm; B: <1.68 – >1.41 mm; C: <2.00 – >1.68 mm; D: ≥2 mm.

**Figure 4 plants-10-00407-f004:**
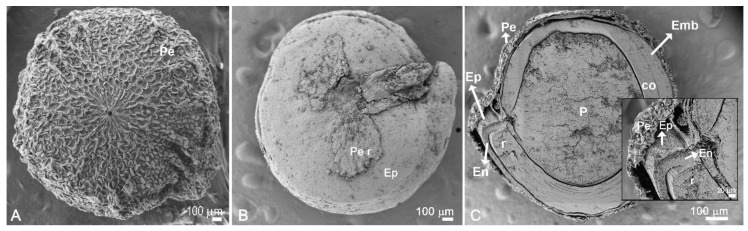
SEM photographs of *C. quinoa* cv. CICA-17. (**A**) grain external view; (**B**) seed with manually removed epicarp; and (**C**) longitudinal medial grain section. Pe, pericarp; Pe r, pericarp partially removed; Ep, episperm; En, endosperm; P, perisperm; Emb, embryo with a radicle-hypocotyl axis (r) and cotyledon (co).

**Figure 5 plants-10-00407-f005:**
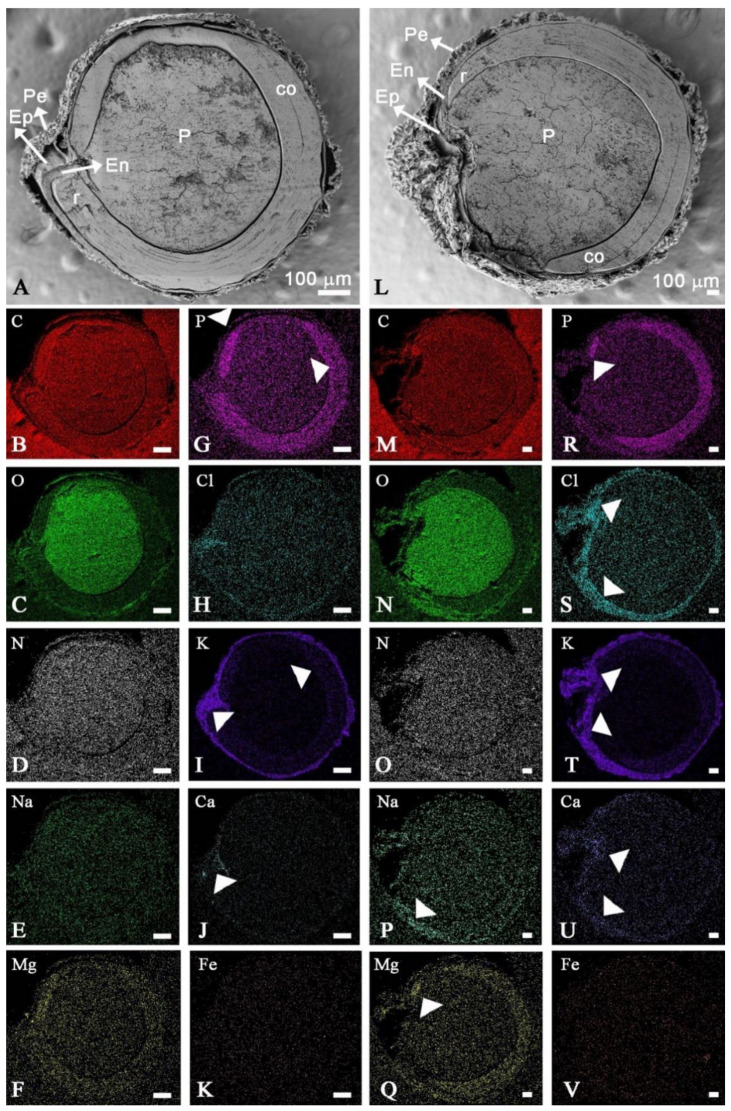
Spatial distribution of elements analyzed by SEM-EDX in grain longitudinal medial sections of *C. quinoa* cv. CICA-17, cultivated under non-saline (**A**–**K**) and saline (**L**–**V**) conditions. Ep, episperm; En, endosperm; P, perisperm; Pe, pericarp; Emb, embryo with the radicle-hypocotyl axis (r) and cotyledon (co). Arrowhead indicates the greater abundance of the mapped element.

**Table 1 plants-10-00407-t001:** Physical and chemical properties of soil samples collected in two saline and non-saline locations of Egypt.

**Physical Properties**										
**depth Soil** **cm**	**pH**	**Sp ^1^** **%**	**EC ^2^** **dS m^−1^**	**DB ^3^** **g cm^−3^**	**OM ^4^** **%**	**CO_3_Ca** **%**	**Sand** **%**	**Silt** **%**	**Clay** **%**	**Textural** **class**
0–60 (non-saline)	8.0	66	2.1	1.51	1.25	2.8	30.0	58.0	12.0	silt loam
0–60 (saline)	8.2	75	26.0	1.92	0.58	0.3	33.3	53.2	3.5	silt loam
	**Chemical Properties**									
	**Na**	**K**	**Ca**	**Mg**	**Cl**	**SO_4_^2^**	**HCO^−3^**	**CO_3_^−2^**	**Fe**	**Mn**
	**meq L^−1^**									
0–60 (non-saline)	7.8	0.6	11.7	4.3	6.5	13.5	4.45	0	4.36	5.29
0–60 (saline)	187.5	6.0	47.0	76.8	280.0	35	1.26	0	5.78	2.44

^1^ Sp: Saturation percentage; ^2^ EC: electrical conductivity; ^3^ DB: bulk density; ^4^ OM: Organic matter. Each value is the mean of n = 6 samples.

**Table 2 plants-10-00407-t002:** Physical and chemical properties of irrigation water of saline and non-saline locations.

Location	EC	pH	Na^+^	K^+^	Ca^2+^	Mg^2+^	Cl^−^	HCO^−3^	CO_3_^−2^	SO_4_^−2^
	dS m^−1^		meq L^−1^							
Non-saline	0.43	7.16	1.04	0.21	1.40	1.22	1.02	0.62	0.00	1.29
Saline	1.70	7.23	6.37	0.35	3.40	4.50	8.50	4.60	0.00	6.12

**Table 3 plants-10-00407-t003:** Mineral content in quinoa grains (pericarps + seeds) under two saline conditions. Significant differences between means (*p* ≤ 0.05) are indicated by different letters behind the values according to the *t*-test.

Element	Non Saline Soil	Saline Soil	Difference (%)	Element	Non Saline Soil	Saline Soil	Difference (%)
	mg kg^−1^ dry weight				mg kg^−1^ dry weight		
**Major elements**				Zr *	0.50 a	0.18 b	64.6
K	9707.62 a	8226.40 b	15.3	Ni	0.48 b	0.30 a	37.3
P	3334.57 b	3959.37 a	18.7	Pb *	0.46 a	0.11 b	75.8
Mg	1443.81 b	1690.27 a	17.1	V *	0.29 a	0.11 b	61.1
Ca	678.22 a	447.04 b	34.1	Ce *	0.09 a	0.05 b	41.8
Na	44.17 b	267.00 a	504.5	As	0.06 a	0.03 b	50.5
**Minor or trace elements**				Ga *	0.06 a	0.02 b	64.9
Fe	72.82 a	49.92 b	31.4	Sn	0.05 a	0.04 b	20.1
Zn *	27.56 a	8.53 b	69.1	Ge *	0.05 a	0.02 b	58.1
Mn	15.30 a	16.19 a	5.9	La *	0.04 a	0.03 b	40.5
Cu	6.70 a	6.34 a	5.3	Li	0.04 b	0.05 a	19.1
Co	0.06 a	0.06 a	2.8	Nb *	0.04 a	0.03 b	34.2
**Ultratrace elements**				Y *	0.04 a	0.02 b	39.0
Si	9968.95 a	7837.06 b	21.4	Nd *	0.04 a	0.03 b	322.0
Al	36.85 a	26.72 b	27.5	Cr	0.29 a	0.16 b	45.3
Ti *	8.61 a	4.18 b	51.5	Pr *	0.01 a	0.01 b	33.5
Sr	3.99 b	7.71 a	92.9	Th *	0.01 a	0.01 b	26.6
Ba *	2.07 a	0.53 b	74.1	Sm *	0.01 a	0.01 b	35.8
Rb *	0.96 b	1.42 a	47.2				

*** Element detected for the first time in quinoa.

**Table 4 plants-10-00407-t004:** SEM-EDX comparative mineral relative percentages on quinoa grain tissues cultivated under non-saline and saline conditions (mean ± SD; n.d., not detected). Significant differences between means (*p* ≤ 0.05) are indicated by different letters behind the values according to the *t*-test. Comparisons were made between identical tissues under the two salinity conditions.

Mineral	Pericarp		Embryo Cotyledon Mesophyll		Endosperm			Perisperm		
	Non-Saline	Saline	Non-Saline	Saline		Non-Saline	Saline		Non-Saline	Saline
C	51.89 ± 3.2 a	50.5 ± 3.7 a	61.32 ± 5.70 a	57.37 ± 2.40 a		61.64 ± 1.40 a	66.17 ± 5.3 a		68.60 ± 2.7 a	74.23 ± 7.10 a
O	39.96 ± 1.5 a	41.4 ± 1.00 a	26.51 ± 5.00 a	30.70 ± 1.40 a		25.76 ± 1.40 a	22.41 ± 3.6 a		26.30 ± 2.1 a	23.89 ± 4.90 a
N	n.d.	n.d.	9.75 ± 0.30 a	10.53 ± 0.10 a		11.46 ± 0.05 a	7.31 ± 1.3 b		4.97 ± 0.01 a	2.50 ± 0.70 b
Na	0.08 ± 0.04 b	0.5 ± 0.03 a	n.d.	n.d.		n.d.	N.d.		n.d.	n.d.
Mg	0.23 ± 0.10 a	0.34 ± 0.10 a	0.38 ± 0.07 a	0.39 ± 0.04 a		0.16 ± 0.03 b	0.38 ± 0.04 a		0.01 ± 0.01 a	0.02 ± 0.01 a
Si	0.26 ± 0.10 a	0.21 ± 0.10 b	n.d.	n.d.		n.d.	n.d.		n.d.	n.d.
P	0.06 ± 0.01 b	0.08 ± 0.04 a	1.01 ± 0.07 a	0.65 ± 0.10 b		0.17 ± 0.01 b	1.23 ± 0.05 a		n.d.	n.d.
S	0.19 ± 0.07 a	0.22 ± 0.2 a	0.45 ± 0.07 a	0.15 ± 0.01 b		0.11 ± 0.01 b	0.69 ± 0.1 a		0.02 ± 0.01 a	0.06 ± 0.01 a
Cl	0.60 ± 0.10 b	1.24 ± 0.40 a	n.d.	0.02 ± 0.01 a		0.07 ± 0.02 a	n.d. b		0.08 ± 0.03 b	0.30 ± 0.20 a
K	6.18 ± 10 a	5.46 ± 3.20 a	0.59 ± 0.05 a	0.22 ± 0.01 b		0.52 ± 0.07 b	1.31 ± 0.10 a		0.12 ± 0.10 b	0.26 ± 0.05 a
Ca	0.31 ± 0.20 a	0.16 ± 0.10 b	n.d.	n.d.		0.12 ± 0.01 a	0.04 ± 0.01 b		n.d.	n.d.
Fe	0.08 ± 0.05 a	0.06 ± 0.04 b	n.d.	0.01 ± 0.00 a		n.d.	n.d.		n.d.	n.d.
Br	0.09 ± 0.09 b	0.13 ± 0.10 a	n.d.	n.d.		n.d.	n.d.		n.d.	n.d.
Al	0.05 ± 0.04 a	0.03 ± 0.03 b	n.d.	n.d.		n.d.	n.d.		n.d.	n.d.
Cu	0.05 ± 0.04 a	N.d. b	n.d.	n.d.		n.d.	n.d.		n.d.	n.d.

## Data Availability

The data presented in this study are available in this article.

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
