# Peer review of "A Long Journey of CICA-17 Quinoa Variety to Salinity Conditions in Egypt: Mineral Concentration in the Seeds"

_plants, 2021, doi:10.3390/plants10020407_

Round 1

Reviewer 1 Report

The manuscript presents the results of a study aimed to evaluate the effect of salinity on the yield and on content of mineral elements and their presence in the various parts of the grain, in a variety of quinoa. The text needs to be revised simplifying and correcting some statements.

More suggestions are provided below and inserted directly in the attached text.

Abstract - The text has not been modified from the previous version. It should be improved in style as there are many unnecessarily repeated expressions.

Introduction – Some suggestions are given in the text

Results – Pay attention to the description of the results. Some suggestions are given in the text

Discussion: Many suggestions are added in the text.

In particular, at lines 211-212, 26 dS/m is the starting value of the electrical conductivity (EC) of the saline soil. Using irrigation water with a higher EC in the environment with saline soil, it can be assumed that the EC is further increased during cultivation. Since the EC value was not measured during and at the end of the crop cycle, it can be supposed that the seeds were produced here under condition of greater saline stress than the starting one.  This is an important criticality of research and here you can take the opportunity to assert it yourself so opening a perspective for further study insights.

Equally, it could be emphasized in this section and/or in the Introduction that although the mineral content is not the only parameter by which to evaluate the nutritional value of a crop (other determinations in this sense have not been made), they still have their own importance, both from this point of view and for the decontamination of soils. This is more true for those determined for the first time in quinoa grains. Also, an explanation of why they have been determined in the various organs of the seed is necessary otherwise this remains a speculative determination only.

Materials and Methods:

Lines 259-260 – delete here and move in Introduction (Line 94)

The paragraph 4.3 can be deleted since the reference to the Tables 3 and 4 has already been done previously and the values of the different parameters are shown in the Tables.

Table 3 – I repeat: I think it is better to clearly distinguish the two parts of the table as "Physical (a) and chemical (b) properties..." Then, add (a) and (b) above the relevant parts of the Table 3.

Conclusion: the text must be simplified giving more space to future perspectives

Lines 339-340 - I repeat: this is uncorrected as only one genotype was studied

Lines 340-349 – the text must be very simplified and expressed correctly.

Lines 351-356 – these statements are more suitable for discussion.

References: Again: some references must be corrected according to the Instructions for authors.

Reviewer 2 Report

I have included my comments in the annotated pdf, which I have attached, to assist the authors in improving the manuscript. The content in Section 4.3 must be moved to the results section and replaced by a full description of the methods used to obtain the data presented in these tables.

Round 2

Reviewer 1 Report

The text needs to be revised clarifying and correcting further some statements. More suggestions are provided directly in the attached text.

Author Response

This manuscript is a resubmission of an earlier submission. The following is a list of the peer review reports and author responses from that submission.

Round 1

Reviewer 1 Report

This manuscript presents an interesting data set on elemental composition of Chenopodium quinoa, an edible plant, in relation to different edaphic conditions. Therefore, it is appropriate study for this journal.

However, there are major issues in the methodology and in the presentation that should be assessed before considering for publication.

Regarding the methodology:

  1. Elemental composition is studied by ICP-MS technique. How were the samples digested? Although there is a reference, at least a simplified protocol should be explained. What kind of analysis was used, quantitative or semiquantitative? This is quite important: digestions of the samples and the type of method selected affect to the precision of the measure. Authors include values of Si, Al and P. An accurate measure of these three elements is difficult to obtain and usually requires specific digestions even with HF acid, and quantitative measures. So, are all the measures provided accurate? Is there any reference material? Which is the error of the measures? In this regard I suggest a revision of the paper: https://doi.org/10.1007/s12011-011-9140-8
  2. The study of elemental distribution uses SEM microscopy coupled to EDX technique. There is no comment in the methodology section of the sample preparation for the SEM. The images provided do not allow a clear distinction between some close tissues, like endosperms and episperm. EDX have a margin of error when an area is selected for analysis. Composition of the areas close to the one selected could affect in the analysis if the surface of the sample is not flat enough. This could be an issue in very close and thin tissues as the ones previously mentioned. In this regard a mapping of transversal and longitudinal sections of saline and non-saline samples could be more informative. See: DOI: 1271/bbb.68.231. EDX results are semiquantitative values and this should be considered. Have you found any crystal or sign of biomineralization in the pericarp or in the testa?
  3. Statistical analysis: how many samples were analyzed? Include the number of samples analyze in the methodology and in the tables and/or figure legends.

Other questions:

Was the irrigation water prepare? How?

Is there only one measure of the soil?

I think organs and tissues are confused through the manuscripts. You only study one organ: the seed. The seed contains several tissues and the embryo with its own tissues.

There are several errors and mistakes in the text.

Reviewer 2 Report

The work is of interest to studies on salinity in relation to plant responses to this stress. Although the work is basic but sufficiently described, the use of the English language is very poor and as it stands cannot be published until this is corrected.

Reviewer 3 Report

The current work presents the nutrient accumulation in quinoa grains parts

The introduction is complete, the material and methods could be improved and provide more details of the relevant methods used. Statistical analysis is missing in some cases. Innovation of the studying is lacking, as salinity effects on quinoa has been extensively studied (even by the same authors). Based on the results, discussion is ok, however, results need to be improved, with additional measurements and critical discussion thereafter. What are the mechanisms involved for the salinity tolerance of the cultivar, what is the physiological behavior of the stressed plants (enzymatic/non enzymatic metabolisms, involved pathways……)

To my opinion the present work needs substancial improvements

  • The new part of this study is the mineral composition/accumulation in different grain parts for a specific cultivar; however, experiment was done once, in specific salinity conditions. some parts of the results are not in agreement with previous relevant reposts on the same species.
  • There are numerus studies that compare different cultivars, different salinity levels, (and abiotic stress), field and hydroponic studies, and in the majority of those studies (some are even from the same authors), the studies goes in depth, examining the effects on nutritive value, on plant photosynthesis, on amino acids content, on antioxidants etc. Some also examined nutrient content in plant tissue and/or grains.
  • Author claimed that for the very first time, quinoa is analysed for the trace elements. This is not true. In Prado et al., 2014, it was reported cobalt, lithium, chromium, selenium, vanadium, molybdenum, zinc, etc.
  • What is the innovation of the present study comparing with the authors’ previous one? ‘’Chenopodium quinoa Willd. A new cash crop halophyte for saline regions of Egypt’’ Another cultivar? In that study, also different seed parts were examined for nutrient content.

Some part of the study are not in Orsini et al., 2011?

  • How do authors support the importance of this study, to examine the minerals of a specific variety, when Bhargava et al., 2008, stated the genetic diversity for mineral accumulation for 40 accessions?
  • CICA, has been studied in Peru. The CICA-127, CICA-17. Is that difference from yours one?
  • In several cases, units are in wrong format and superscript is missing
  • Tables 1 & 2. Statistical analysis-differences and replications are missing. Is it one measurement per soil?
  • Figure 3. Add statistical letters to show the significant differences
  • Table 4. The analysis related to that table is in many cases wrong. Firstly, it is not stated by the authors what is compared with what. Are the significant differences reflected to each individual grain part in saline and non-saline conditions? Is the comparison made also within all parts?
    • Based on N analysis, I understand that it is compared the N in perisperm for both saline and no saline conditions. If this is the way of analysis, several other cases are wrong
    • P increased in pericarp in saline: this is not supported by the statistics. Why you have ‘’b’’ in phosphorus endospem in saline, if P is increased?
    • Same for K

Some suggestions/comments

  • Authors mentioned that grain had main elements. Isn’t it obvious that?
  • Did author measure amino acid content, for to include that aspect at the conclusion sentence of an abstract?
  • Iron is not a macronutrient
  • L131-132. The way of presentation is not correct. For example ‘’≥1.41’’ means ‘’greater than ‘’ not ‘’greater than 1.41 but less than 1.68’’ as authors wanted to say-present
  • This is not true. In saline, the 1.41 mm is less than in non-saline conditions
  • What was the composition and nutritional value of the large grain. This would provide info to the present study.
  • Delete the year of publication
  • Delete ‘’and’’ after ‘’were’’

Reviewer 4 Report

The manuscript presents the results of the effect of salinity on the yield and content of mineral elements and their presence in the various parts of the grain in a variety of quinoa. The text is often badly organized and needs to be revised simplifying some statements which are often repetitive and messy. English revision by a native speaker is also required. More suggestions are provided below and inserted directly in the attached text.

Introduction: The text is repetitive and messy. More linearity is needed to facilitate reading and understanding. Concepts must be expressed concisely according to the purpose of the research. Rewrite better.

Results:

2.1: this subsection must be moved to Materials and Methods

2.4: simplify this sub-section

2.5: write this sub-section better as it is not well organized and it becomes very difficult to highlight the most significant results

In 4.6 and in all Tables and Figures the statistical tests used are not the most appropriate ones

Discussion: the text must be simplified and ordered, focusing on the most important results: effect on yield and explanation of discrepancies with the 1000-seed weight and with the results reported in the literature; importance of the presence of some mineral elements, especially those with greater biological value, and of the effects of salinity on their content; importance of the saponification during the processing of quinoa grains and its effect on the content of some elements that have greater nutritional value; importance of some elements present in trace. The compartmentalization of some minerals in the roots and vegetative organs of the plant is a fact not directly detected in this study and, therefore, their role should be described in general. The References need to be more appropriately integrated with the discussion of results

Conclusion: the text must be simplified giving more space to future perspectives. Some statements are more suitable for discussion.

References: some references must be corrected according to the Instructions for authors
